# Novel *Babesia bovis* exported proteins that modify properties of infected red blood cells

**Hassan Hakimi**[1,2]*, **Thomas J. Templeton**[1], **Miako Sakaguchi**[3], **Junya Yamagishi**[4,5], **Shinya Miyazaki**[1], **Kazuhide Yahata**[1], **Takayuki Uchihashi**[6], **Shin-ichiro Kawazu**[2], **Osamu Kaneko**[1], **Masahito Asada**[1,2]*

1 Department of Protozoology, Institute of Tropical Medicine (NEKKEN), Nagasaki University, Nagasaki, Japan, 2 National Research Center for Protozoan Diseases, Obihiro University of Agriculture and Veterinary Medicine, Obihiro, Hokkaido, Japan, 3 Central Laboratory, Institute of Tropical Medicine (NEKKEN), Nagasaki University, Nagasaki, Japan, 4 Division of Collaboration and Education, Research Center for Zoonosis Control, Hokkaido University, Sapporo, Japan, 5 International Collaboration Unit, Research Center for Zoonosis Control, Hokkaido University, Sapporo, Japan, 6 Department of Physics, Nagoya University, Aichi, Japan

* hhakimi@obihiro.ac.jp (HH); masada@obihiro.ac.jp (MA)

**Data Availability Statement:** All relevant data are within the manuscript and its Supporting Information files.

## Abstract

*Babesia bovis* causes a pathogenic form of babesiosis in cattle. Following invasion of red blood cells (RBCs) the parasite extensively modifies host cell structural and mechanical properties via the export of numerous proteins. Despite their crucial role in virulence and pathogenesis, such proteins have not been comprehensively characterized in *B. bovis*. Here we describe the surface biotinylation of infected RBCs (iRBCs), followed by proteomic analysis. We describe a multigene family (*mtm*) that encodes predicted multi-transmembrane integral membrane proteins which are exported and expressed on the surface of iRBCs. One *mtm* gene was downregulated in blasticidin-S (BS) resistant parasites, suggesting an association with BS uptake. Induced knockdown of a novel exported protein encoded by BBOV_III004280, named VESA export-associated protein (BbVEAP), resulted in a decreased growth rate, reduced RBC surface ridge numbers, mis-localized VESA1, and abrogated cytoadhesion to endothelial cells, suggesting that BbVEAP is a novel virulence factor for *B. bovis*.

## Author summary

*Babesia bovis* is an apicomplexan intraerythrocytic protozoan parasite which causes the most pathogenic form of babesiosis in cattle. Like other apicomplexan parasites, *B. bovis*-induced modification of host cells is crucial for its survival. However, our knowledge of *Babesia* surface exposed proteins is limited to variant erythrocyte surface antigen1 (VESA1), which is responsible for fatal cerebral babesiosis. Here we identified two novel exported proteins in *B. bovis* using red blood cell (RBC) surface biotinylation and mass spectrometry. One of the proteins was determined to be essential for parasite development and pathogenicity. Induced knockdown of this protein resulted in a decreased growth rate, reduced RBC surface protrusions created by the parasite, mis-localized VESA1, and

**Funding:** This study was supported partly by grants from Japan Society for the Promotion of Science (https://www.jsps.go.jp/english/) to H.H. (15K18783, 19K15983), M.A. (16K08021, 19K06384), S.K. (18K19258, 19H03120) and O.K. (16F16105). This work was supported by NRCPD OUAVM Joint Research Grant of NRCPD, Obihiro University of Agriculture and Veterinary Medicine (https://www.obihiro.ac.jp/facility/protozoa/ento) to M.A. (28-11, 29-2, 30-1). H.H. is a recipient of the JSPS Postdoctoral Fellowship for foreign researchers from the Japan Society for the Promotion of Science. T.J.T. was supported by a visiting professorship to the Institute of Tropical Medicine, Nagasaki University (http://www.tm.nagasaki-u.ac.jp/nekken/en/). The funders had no role in study design, data collection and analysis, decision to publish, or preparation of the manuscript.

**Competing interests:** The authors have declared that no competing interests exist.

abrogated cytoadhesion to endothelial cells. VESA1 is a ligand for cytoadhesion of iRBCs to capillary endothelial cells which leads to blockage of capillaries and causes cerebral symptoms. The second identified protein is encoded by a large multigene family. The gene was downregulated in blasticidin-S resistant parasites, suggesting that the protein mediates entry of blasticidin-S and likely other solutes across the iRBC membrane. This is the first description of a putative channel or transporter molecule on the surface of *Babesia*-iRBC. Our results provide new insights into the host cell modifications by *B. bovis* and their pathogenicity.

## Introduction

Babesiosis is an emerging tick-borne disease affecting animals and humans which is caused by intraerythrocytic protozoans of the genus *Babesia*. The parasite infects a wide range of vertebrates and causes great economic losses in livestock. The burden of bovine babesiosis in tropical and subtropical regions is attributed to *Babesia bovis* and *Babesia bigemina*. While pathogenesis of *B. bigemina* is mainly related to intravascular hemolysis, sequestration of *B. bovis*-infected red blood cells (iRBCs) in internal organs and brain produces severe clinical symptoms which occasionally result in fatality [1]. Constraints against disease control include the low efficacy of available live vaccines for *B. bovis*, limited treatment options, and the emergence of drug and acaricide resistance of tick vectors [1–3]. The combination of new drugs and vaccine intervention are required to better control the disease.

Following RBC invasion of merozoites or sporozoites injected by ticks, *B. bovis* parasites modify the host cell via the export of numerous proteins to the RBC cytoplasm and surface, to facilitate metabolite exchange, increase RBC rigidity, and to mediate cytoadherence in deep tissues [4–6]. Modification of iRBCs results in the production of unique surface protrusions, ridges, which are the focal points for adhesion to endothelial cells [4,7,8]. Cytoadherence causes sequestration of iRBCs in the microvasculature of internal organs, thus avoiding spleen clearance. The binding of iRBCs to unknown receptor(s) on brain microvascular endothelial cells can cause cessation of blood flow and produce cerebral symptoms [8].

Although the mechanisms of protein export to the RBC cytoplasm and surface are not known, several exported proteins have been reported in *B. bovis* [9–15]. The majority of known exported proteins are the products of multigene families such as variant erythrocyte surface antigen 1 (VESA1), small open reading frame proteins (SmORFs), and spherical body protein 2 (SBP2). VESA1 proteins are heterodimeric proteins encoded by the largest multigene family, *ves1*, in *B. bovis* [16]. They cluster on the surface of ridges, undergo antigenic variation, and are responsible for host immune evasion and cytoadhesion [4,11]. SmORFs are produced from the second largest gene family, *smorfs*, and are exported to the RBC during parasite development [15,17]. While the function of SmORFs is unknown, their gene distribution and proximity to *ves1* genes throughout the *B. bovis* genome indicates a role in VESA1 biology [17]. SBP2 is also encoded by a multigene family, *sbp2*. Unlike *ves1* and *smorfs*, which are unique to *B. bovis*, *sbp2* are conserved across the genus *Babesia* [17–19]. SBP2 are localized to spherical bodies, organelles analogous to dense granules in other apicomplexan parasites, and are released into the RBC cytoplasm upon invasion [9]. To date SBP1, SBP2, SBP3, and SBP4 have been characterized in *B. bovis* [9,10,12,13] but their functions during invasion and development in the RBC are unknown. Recently, Gohil et al. (2013) identified three novel exported proteins of *B. bovis* among 214 putative exported protein annotated for the presence of a signal peptide cleavage site but negative for transmembrane domains (TM) or glycosylphosphatidylinositol (GPI) anchor attachment motif [14]. Pellé et al. (2015) further

refined this list to include 59 proteins harboring a PEXEL-like motif (PLM) preceded by a signal peptide sequence [15]. PEXEL (*Plasmodium* export element) is a unique conserved pentameric motif at the N-terminal of approximately 300 proteins in *Plasmodium* [20,21] which is cleaved within the endoplasmic reticulum by a protease and determines protein export [22,23]. A similar signal-mediated pathway was identified in *Toxoplasma gondii*, but as a sorting signal to dense granules before release into the parasitophorous vacuole (PV) space [24].

Despite their crucial role in *B. bovis* virulence and pathogenesis, iRBC surface exposed proteins have not been comprehensively characterized. In this study we performed biotinylation of iRBC surface proteins, allowing their extraction, purification, and proteomic analysis. We confirmed export of several candidate proteins and performed initial protein characterizations, including a family of multi-transmembrane integral membrane proteins and a novel virulence factor associated with iRBC cytoadhesion to endothelial cells.

## Results

### RBC surface proteomics

Biotinylation coupled to liquid chromatography-tandem mass spectrometry (LC-MS/MS) was used to obtain the first comprehensive surface proteome of *B. bovis*-iRBCs. We selected *B. bovis* for adhesion to bovine brain endothelial cells (BBECs) by panning uncloned parasites and the resulting cytoadherent parasite line was used for biotinylation (S1A Fig). The surface proteins of enriched iRBCs from *in vitro* cultured *B. bovis* were biotinylated, extracted, purified, and analyzed by LC-MS/MS in three biological replicates. Surface protein biotinylation was confirmed by live fluorescence microscopy. Biotinylation did not affect the integrity of iRBC membranes and the release of merozoites (S1B Fig). The extraction and purification of biotinylated proteins was confirmed by Western blot analysis (Fig 1A). A complete list of *B. bovis* proteins exported from the Scaffold software is provided in the supplemental

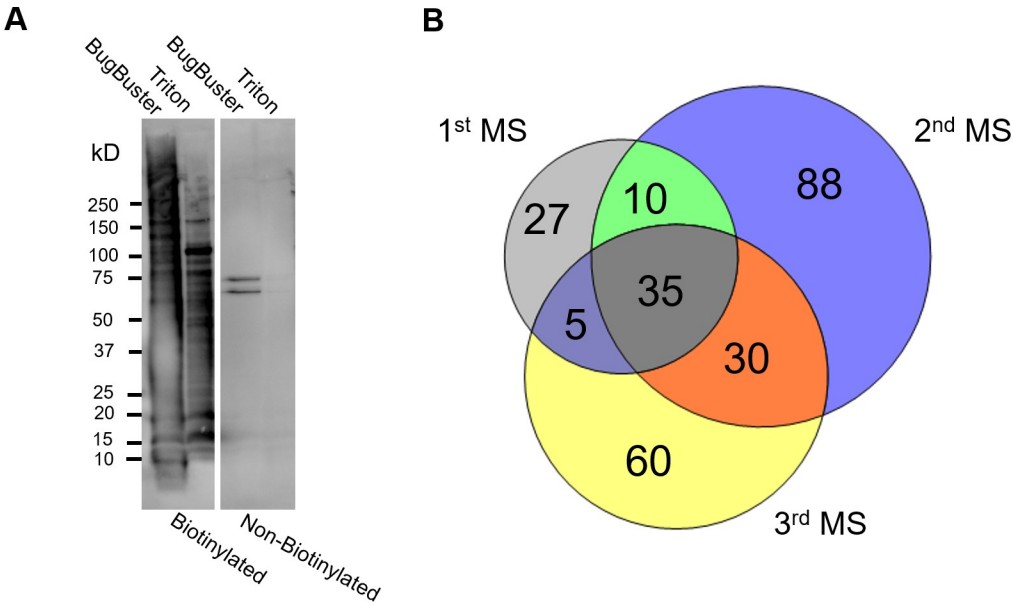

**Fig 1. Biotinylation and liquid chromatography-tandem mass spectrometry (LC-MS/MS) of *B. bovis*-iRBCs. (A)** Western blot analysis of sequentially extracted proteins from biotinylated and control (non-biotinylated) samples. The image is representative of three independent experiments done with an approximately two-month interval. The membrane was probed with horseradish peroxidase (HRP)-conjugated streptavidin. **(B)** Venn diagram showing the number of *B. bovis* proteins identified from biotinylated samples by three independent LC-MS/MS analyses.

information (S1 Table). In the first attempt, we lysed the iRBCs by a hypotonic solution followed by detergent protein extraction using the BugBuster reagent, which was shown to be efficient for *P. falciparum*-iRBCs surface proteins such as PfEMP1 [25]. LC-MS/MS analysis showed bovine hemoglobin contamination which might interfere with identification of *B. bovis* peptides. Therefore, in the second and third attempts, protein extraction was done following biotinylation and saponin treatment. This modification improved the detection of total *B. bovis* peptides and proteins. Mass spectra obtained from LC-MS/MS were searched against *B. bovis* and *Bos taurus* databases. Because parasite-encoded surface exposed proteins are low in abundance, we allowed protein identification by a single peptide. The number of unique peptides identified in the first, second, and third attempts from biotinylated samples were 805, 1600, and 948, respectively, in which 24–26% of identified peptides were from *B. bovis* and yielded 77, 163, and 130 proteins, respectively (Table 1). Out of the 255 unique *B. bovis* protein hits, 80 proteins detected from biotinylated samples by at least two MS analyses (Fig 1B) were further annotated for predicted export based upon the presence of a signal sequence, a transmembrane (TM) domain, or GPI anchor. This refinement resulted in a total of 38 putative secretory proteins of which 20 had no predicted function in the database (S2 Table). The well-known surface exposed *B. bovis* protein, VESA1, was identified in all three MS. VESA1 lacks an N-terminal signal sequence but has a C-terminal TM domain [11]. One SmORF and one SBP2, the products of two multigene families in *B. bovis*, SBP1 and SBP3 were also detected. These proteins have an N-terminal signal sequence and were shown to be exported into the iRBC cytoplasm [9,10,12,13,15]. Of the 38 identified proteins 32 possessed a PEXEL-like export motif (PLM; RxL or RxxL) proposed by Pellé et al. [15].

## Validation of localization for a subset of candidates

To evaluate the localization of putative exported proteins, we transformed parasites with plasmid constructs expressing target molecules fused with 2 myc epitopes (S2A Fig). We selected 10 proteins from the candidate list based on the three criteria for secretion and their abundance in the MS analysis (S3 Table) and further evaluated together with two positive control proteins, SBP3 and a variant of SmORF. Indirect immunofluorescence antibody test (IFAT) revealed that signals of three proteins were detected in iRBCs among selected 10 candidates. Two proteins showed signals inside the parasite and the edge of the iRBCs like SBP3 and SmORF, namely, BBOV_III000060 (Bb60-mtm) and BBOV_III011920 (Bb11920-mtm) (Fig 2A). These proteins possess ten TM domains each and are paralogs encoded by a multigene family (termed *multi-transmembrane (mtm)* family). Each possess an RxL motif at amino acid positions 24–27. The

**Table 1. Summary of identified proteins.**

| | 1st MS | | | | 2nd MS | | | | 3rd MS | | | |
|---|---|---|---|---|---|---|---|---|---|---|---|---|
| | Biotinylated | | Non-biotinylated | | Biotinylated | | Non-biotinylated | | Biotinylated | | Non-biotinylated | |
| | Peptides* | % | Peptides* | % | Peptides | % | Peptides | % | Peptides | % | Peptides | % |
| *Bos taurus* | 612 | 76.0 | 179 | 89.9 | 1184 | 74.0 | 261 | 89.7 | 701 | 74.3 | 227 | 83.2 |
| *Babesia bovis* | 193 | 24.0 | 20 | 10.1 | 416 | 26.0 | 30 | 10.3 | 247 | 26.2 | 46 | 16.8 |
| Total | 805 | | 199 | | 1600 | | 291 | | 948 | | 273 | |
| | Proteins | % | Proteins | % | Proteins | % | Proteins | % | Proteins | % | Proteins | % |
| *Bos taurus* | 255 | 76.8 | 114 | 82.0 | 389 | 70.5 | 119 | 79.9 | 259 | 66.6 | 97 | 68.8 |
| *Babesia bovis* | 77 | 23.2 | 25 | 18.0 | 163 | 29.5 | 30 | 20.1 | 130 | 33.4 | 44 | 31.2 |
| Total | 332 | | 139 | | 552 | | 149 | | 389 | | 141 | |

* The peptide numbers are from BugBuster protein extract

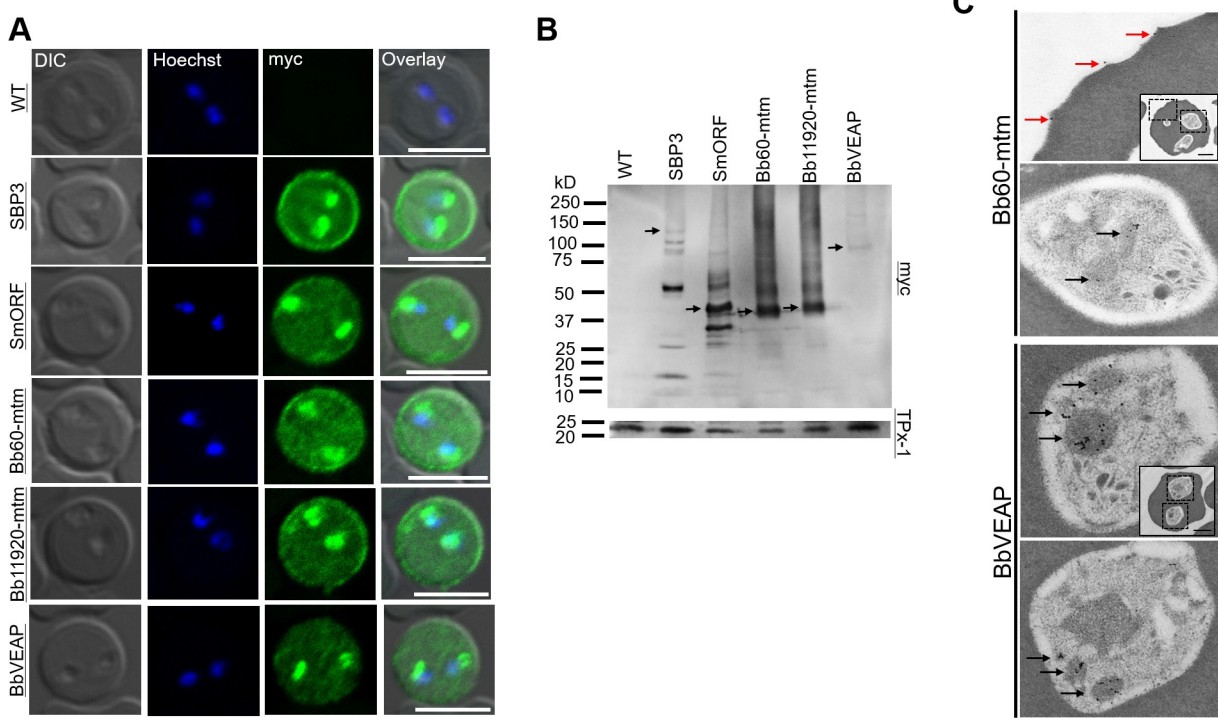

**Fig 2. Expression and localization analysis of candidate proteins determined by indirect immunofluorescence antibody test (IFAT), Western blotting, and immunoelectron microscopy (IEM).** **(A)** IFAT of parental wild type (WT) and transgenic *B. bovis* lines expressing myc-tagged target proteins. The parasites were reacted with anti-myc antibody (α-myc, green) and nuclei were stained with Hoechst 33342 (Hoechst, blue). Scale bar = 5 μm. **(B)** Western blot analysis of transgenic *B. bovis* expressing myc-tagged proteins and WT parasite (WT). The expected full-length bands of the proteins are indicated with black arrows. **(C)** Immunoelectron microscopic analysis of transgenic *B. bovis* expressing Bb60-mtm or BbVEAP tagged with myc epitopes. Anti-myc antibody shows concentration of Bb60-mtm and BbVEAP in spherical bodies (black arrows) and Bb60-mtm expression on the iRBC surface (red arrows). Scale bar = 1 μm.

third protein, BBOV_III004280 (BbVEAP in the Fig 2A) showed signals inside iRBCs and parasites, but not from the edge of the RBC (Fig 2A). The deduced amino acid sequence of BBOV_III004280 possesses a signal peptide with an RxL motif at amino acid position 185–188 and is conserved among all piroplasms except *B. microti*. We designated BBOV_III004280 as VESA export-associated protein (VEAP) gene because knockdown of this gene suppressed VESA export as described below. The remaining seven candidates did not show signals in iRBCs by IFAT, suggesting that they were not exported to iRBCs (S2B Fig). Thus, we decided to further characterize three novel proteins that were exported to iRBC. Target protein expression in transgenic parasites was confirmed by Western blotting (Fig 2B). SBP3 and SmORF showed several bands indicating possible processing during export or degradation during protein extract preparation [15]. The products of the *mtm* genes Bb60-mtm and Bb11920-mtm showed bands at the expected size of 49 kDa plus band smears. BbVEAP showed a single band corresponding to the anticipated size of 100 kDa. Immunoelectron microcopy (IEM) revealed that Bb60-mtm is localized in the spherical bodies of merozoites and on the iRBC surface close to ridge structures (Fig 2C, S3 Fig). BbVEAP was detected in merozoite spherical bodies (Fig 2C, S3 Fig).

## A multigene family encoding Bbmtm is expanded in *B. bovis*

Of the three novel proteins that were exported into iRBC, Bb60-mtm and Bb11920-mtm are paralogous proteins with similar architectures of ten TM domains each and molecular weights

of approximately 49 kDa. Via BLAST searches of PiroplasmaDB and GenBank, we found 44 total copies in the *B. bovis* genome and named this newly identified expanded gene family *mtm* (Fig 3A, S4 Table). The genes are typically telomeric and located in gene neighborhoods containing *ves1* and *smorf* genes, two gene families encoding important exported proteins in *B. bovis* (Fig 3B). In *P. falciparum*, exported proteins are also typically localized to clusters within telomeric regions [26]. The family *mtm* exists in another *Babesia* species from sheep (*Xinjiang*), which is closely related to *B. bovis* [27], and to date the family is unique to these two species (Table 2). To examine in other Piroplasmida the possible expansion of gene families that encode multi-transmembrane proteins, we used TMHMM-2.0 to screen whole genome and proteome datasets (Table 2, Fig 3C and 3D). Specifically, a major facilitator superfamily (*mfs*)-like gene is greatly expanded in *B. ovata* and *B. bigemina*; while *tpr* (*T. parva* repeat) is significantly expanded in *Theileria* spp. [28,29], though several homologs of *tpr* exist in *Babesia* spp. Homology clustering shows that *tpr* in *T. parva* and *T. annulata* make one cluster while *T. orientalis tpr* forms a separate cluster (Fig 3C). Homology clustering of *mtm*

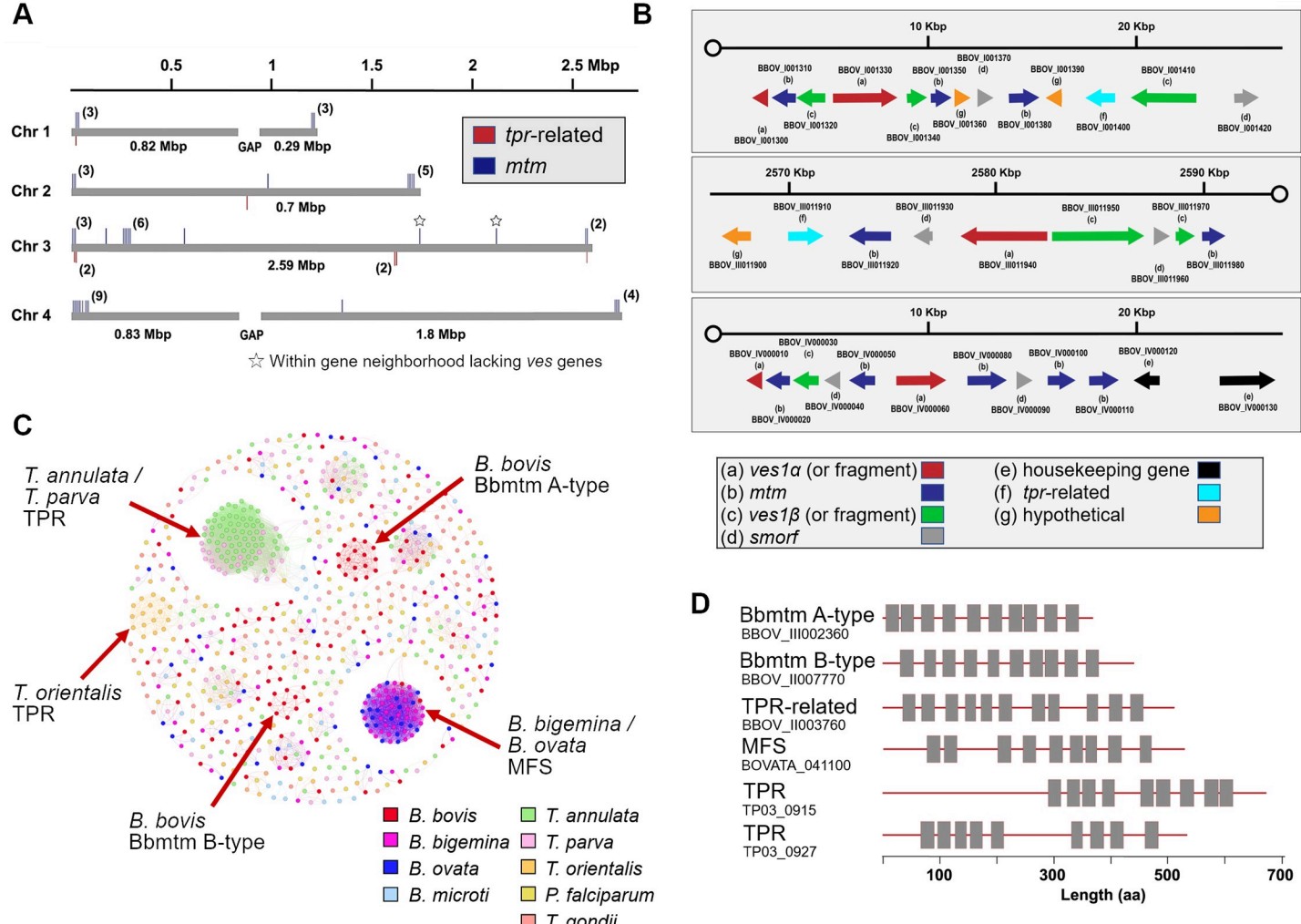

**Fig 3. Expansion and distribution of genes encoding multi-transmembrane proteins in piroplasms.** (**A**) Distribution of *mtm* and *tpr*-related genes in the *B. bovis* genome. (**B**) Arrangement of multigene families (*ves1*, *mtm*, *smorf*, and *tpr*-related genes) in the nuclear genome of *B. bovis*. (**C**) Homology clustering based on sequence similarities of genes with more than eight TM domains in piroplasms, *Plasmodium falciparum*, and *T. gondii*. (**D**) Schematics of selected gene products with multiple TM domains in piroplasms. Box indicates predicted TM domain.

**Table 2. Distribution of *tpr*-related, *mtm* and *mfs* genes in Piroplasmida.**

| Gene family | Species | | | | | | |
|---|---|---|---|---|---|---|---|
| | *B. bovis* | *B. sp. (Xinjiang)* | *B. ovata* | *B. bigemina* | *B. microti* | *T. annulata* | *T. parva* |
| *tpr*-related | 7 | 4 | 3 | 3 | 5 | 50 | 44 |
| *mtm* | 44 | 7–8 | - | - | - | - | - |
| *mfs* | 2 | 2 | 28 | 41 | - | 1 | 1 |

revealed the existence of two clusters for *B. bovis*: A-type and B-type, and Bb60-mtm and Bb11920-mtm belong to B-type (Fig 3C, S4 Table). In summary, although orthologous relationships and expansions of genes encoding homologous multi-transmembrane proteins are not generally conserved across the *Babesia* and *Theileria* genera, what is conserved is a theme of lineage-specific expansions of gene families that encode multi-transmembrane proteins which are candidates for export to the iRBC.

## Bbmtm is associated with BS uptake

The presence of expanded gene families encoding predicted exported multi-transmembrane proteins across piroplasmida suggests that intraerythrocytic development of these parasites requires *de novo* channel or transporter activity across the iRBC membrane. New permeability pathways have been described for the *P. falciparum*-iRBC membrane [30, 31], and the activity of a plasmodial surface anion channel (PSAC) was shown to be determined by the protein products of *P. falciparum clag3.1* and *clag3.2* genes [32] within the expanded *clag* gene family. Orthologs of *clag* genes are absent in piroplasmida, but *Babesia* parasites increase iRBC permeability to several organic solutes including sorbitol, suggesting the existence of channels or transporters [33]. Selection of *P. falciparum* under high BS concentration resulted in silencing of both *clag3* genes, suggesting that resistance to BS happens through epigenetic downregulation of *clag* genes [34]. To gain insights into *mtm* function, we produced two BS-resistant lines of *B. bovis* by exposing the parasites to increasing concentrations of BS. These parasites showed increased half maximal inhibitory concentration ($IC_{50}$) of BS compared to wild type (WT) (9.4 and 10.7 vs 2.3 μg/mL, Fig 4A), and delayed lysis in sorbitol lysis assays compared to WT (Fig 4B). RNA-seq analysis revealed that transcripts of several genes including one *mtm* (*Bb60* for BS-resistant line 1 and BBOV_III000010 (*Bb10*) encoding an A-type Bbmtm for BS-resistant line 2) among many *mtm* family members was less abundant than in WT, suggesting that these *mtms* are potentially linked to BS uptake activity of iRBCs (Fig 4C, S5 Table). The sensitivity to BS was partially reversed in parasites cultured in the absence of BS for approximately two months (5 and 2.9 μg/mL, respectively), suggesting epigenetic regulation of this drug resistance. RNA-seq and qRT-PCR confirmed the recovery of downregulated *Bb60* and *Bb10* in the corresponding revertant parasite lines (Fig 4D, S4 Fig). Episomal overexpression of Bb60-mtm and Bb10-mtm in BS-resistant line 1 and 2 showed fluorescence signals inside the parasite and the edge of the iRBCs as expected (Fig 4E) and made these parasites more sensitive to BS (Fig 4F; $IC_{50}$ of 7.8 and 2.3 μg/mL for BS-res1-Bb60 and BS-res2-Bb10, respectively), supporting our hypothesis on their role in BS uptake. However, episomally overexpressing Bb60-mtm with 10 or 100 nM WR99210 (Bb60-10 or Bb60-100 lines, respectively) in WT BS-sensitive line did not change $IC_{50}$ to BS (2.8 or 2.5 μg/mL, respectively), perhaps indicating saturation of the channel activity (S5 Fig).

## Characterization of BbVEAP

Our three attempts to disrupt the BbVEAP gene locus, the third protein in our list which is exported into iRBC, using CRISPR/Cas9 system were unsuccessful, suggesting possible

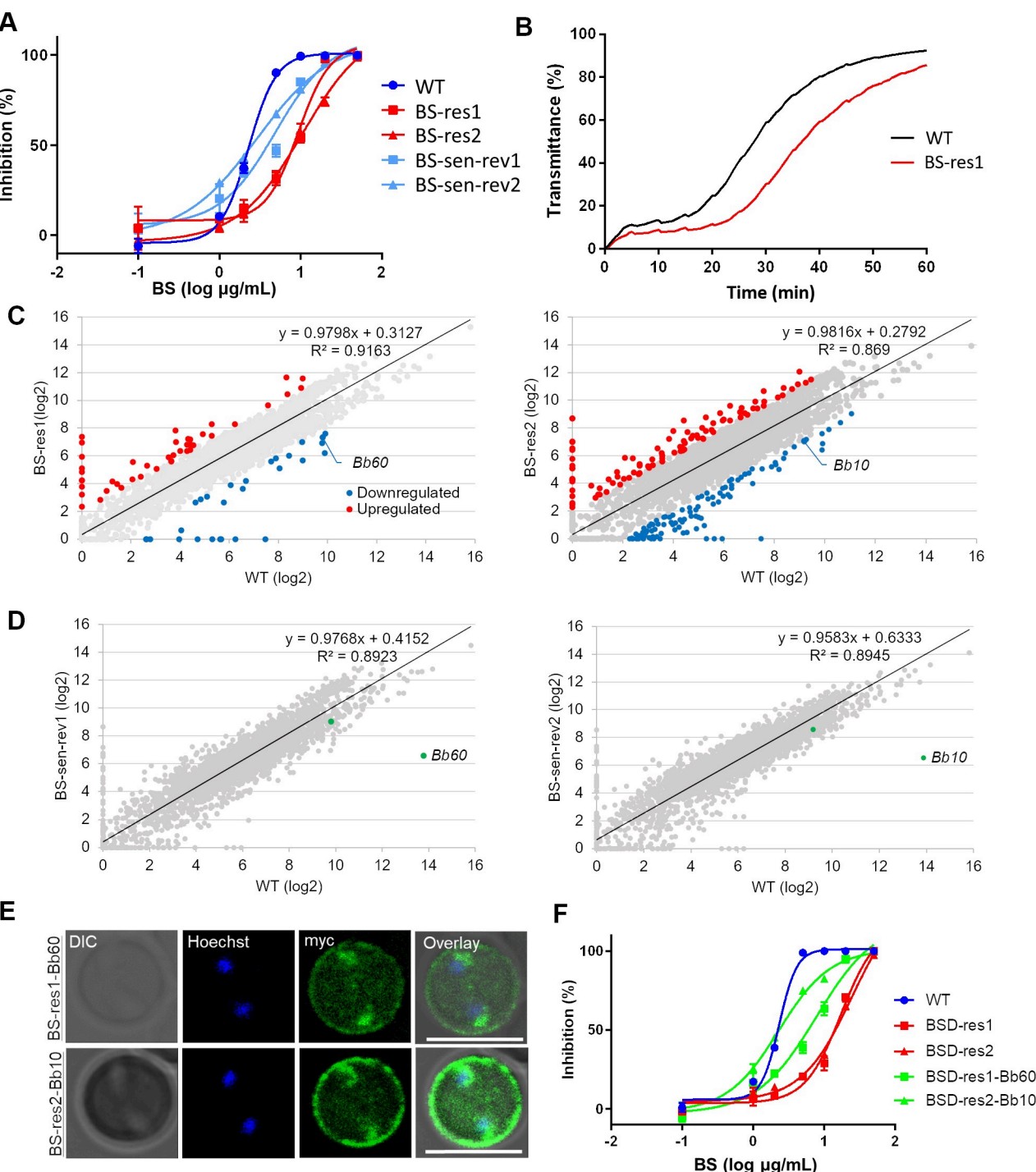

**Fig 4. Blasticidin S-resistance in *B. bovis* is linked with downregulation of *Bb60*.** (**A**) Growth inhibition curves of parasite lines in the presence of different concentrations of BS (μg/mL). All data are expressed as mean ± SEM of triplicate cultures. (**B**) Osmotic lysis of *B. bovis* wild type (WT) and BS-resistant line 1 in the presence of sorbitol. iRBCs were enriched and the lysis experiment was performed at 37˚C. The graph is representative from two biological replicates done within a two-week interval. (**C**) The scatter diagram showing the differential expression of genes in *B. bovis* WT and BS-resistant lines. The horizontal and vertical axes represent the log 2-fold expression of WT and BS-resistant parasites, respectively. The upregulated and downregulated genes are shown in red and blue colors, respectively. (**D**) Scatter diagram showing the differential expression of genes in *B. bovis* WT and BS-sensitive revertant lines. The expression of downregulated *mtms* in BS-resistant lines, *Bb60* and *Bb10*, recovered in revertant lines (green dots). (**E**) Indirect immunofluorescence antibody test of transgenic *B. bovis* BS-resistant lines episomally expressing myc-tagged Bb60-mtm or Bb10-mtm stained with anti-myc (green). The parasite nuclei were stained with Hoechst 33342 (Hoechst, blue). Scale bar = 5 μm. (**F**) Growth inhibition curves of a panel of parasite lines in the presence of different concentrations of BS (μg/mL). Bb60-mtm and Bb10-mtm are episomally overexpressed under 100 nM WR99210 in BS-resistant lines 1 and 2, respectively. All data are expressed as mean ± SEM.

essentiality for the parasite. To functionally characterize BbVEAP we inserted a glucosamine (GlcN)-inducible *glmS* riboswitch together with 2 myc epitopes at the 3' end of the BbVEAP open reading frame (ORF) [35]. Integration of the *glmS* sequence into the endogenous locus was confirmed by PCR (Fig 5A) and the expression of the myc-tagged protein was confirmed by Western blot analysis with the predicted band size (Fig 5A and 5B). In the absence of GlcN, *glmS*-tagged parasites demonstrated a roughly 80% reduction in basal BbVEAP protein expression by Western blot analysis (Fig 5B and 5C). This reduction did not affect the growth of parasites compared to the control (Fig 5D). The reduction of BbVEAP protein expression without addition of GlcN could be due to leakiness of the *glmS* system, decrease of mRNA stability or translation due to *glmS* sequence at its 3' end, or endogenously produced GlcN by *B. bovis* as is reported for *Trypanosoma cruzi* [36], an observation for future verification. Addition of GlcN resulted in a dose dependent parasite growth reduction of BbVEAP-myc-*glmS* lines compared to the control BbVEAP-myc parasite (S6 Fig). Significant reduction of BbVEAP protein expression (82–92% reduction) with 2.5 mM GlcN, the maximum concentration without effect on the control parasite, was confirmed by Western blot analysis (Fig 5B and 5C). This knockdown resulted in a significant decrease in the growth rate accompanied by a significant increase of ring stage (immature) parasites and a decrease of binary form (mature) parasites, suggesting a defect in parasite development (Fig 5D and 5E).

Because BbVEAP is deposited in spherical bodies, we examined whether the knockdown of BbVEAP affected structure of these organelles by transmission electron microcopy (TEM). While no clear changes were seen in spherical body structure, we noticed fewer RBC surface ridges (Fig 5F). Scanning electron microscopy (SEM) images further revealed that the ridges were less protrusive with GlcN treatment in addition to a reduction in number (S7 Fig). To exclude the contribution of parasite stage on ridge numbers, we quantified the ridges in the mature stage of the parasite (binary form) using TEM images. GlcN treatment significantly reduced the number of ridges on the surface of iRBCs (Fig 5G). Because ridges are the focal point for adhesion of *B. bovis*-iRBCs to endothelial cells [8], we examined whether knockdown of BbVEAP affected the cytoadhesion of iRBCs to BBECs. For this purpose, we used a cloned parasite from the cytoadherent *B. bovis* line (S1A Fig) to generated additional transgenic parasite lines in which a myc-*glmS* sequence was inserted within the 3' end of the BbVEAP ORF the same way as the BbVEAP-myc-*glmS* line. The parasites were treated with GlcN for 3 days and cytoadhesion assays were conducted. BbVEAP expression was dramatically reduced in two BbVEAP-knockdown clones with GlcN treatment as determined by Western blot analysis (S8A Fig). While the addition of GlcN had no effect on the binding ability of WT parasites, cytoadhesion was abrogated in two BbVEAP-knockdown clones (Fig 6A). Although the expressed amount of VESA1 appears to be unchanged by Western blotting (S8A Fig), its distribution was affected; parasites without GlcN treatment showed a punctate pattern in the iRBC resembling the expression of VESA1 on ridges while in parasites with GlcN treatment signal was only detectable within the parasite cytoplasm (Fig 6B). Unlike VESA1, the expression and localization of SBP4, another exported protein into iRBC, was not affected when treated with GlcN (Fig 6B, S8A Fig).

## Discussion

In this study we used iRBC surface biotinylation coupled with mass spectrometry to characterize the surface proteome of *B. bovis*. This approach was successfully used to describe the *Plasmodium* parasite surface and iRBC surface parasite-encoded proteome [37,38]. We prioritized 38 proteins for potential export into iRBC, based upon their abundance in the above assay and subsequent annotation for export motifs. This list includes known exported proteins, such as

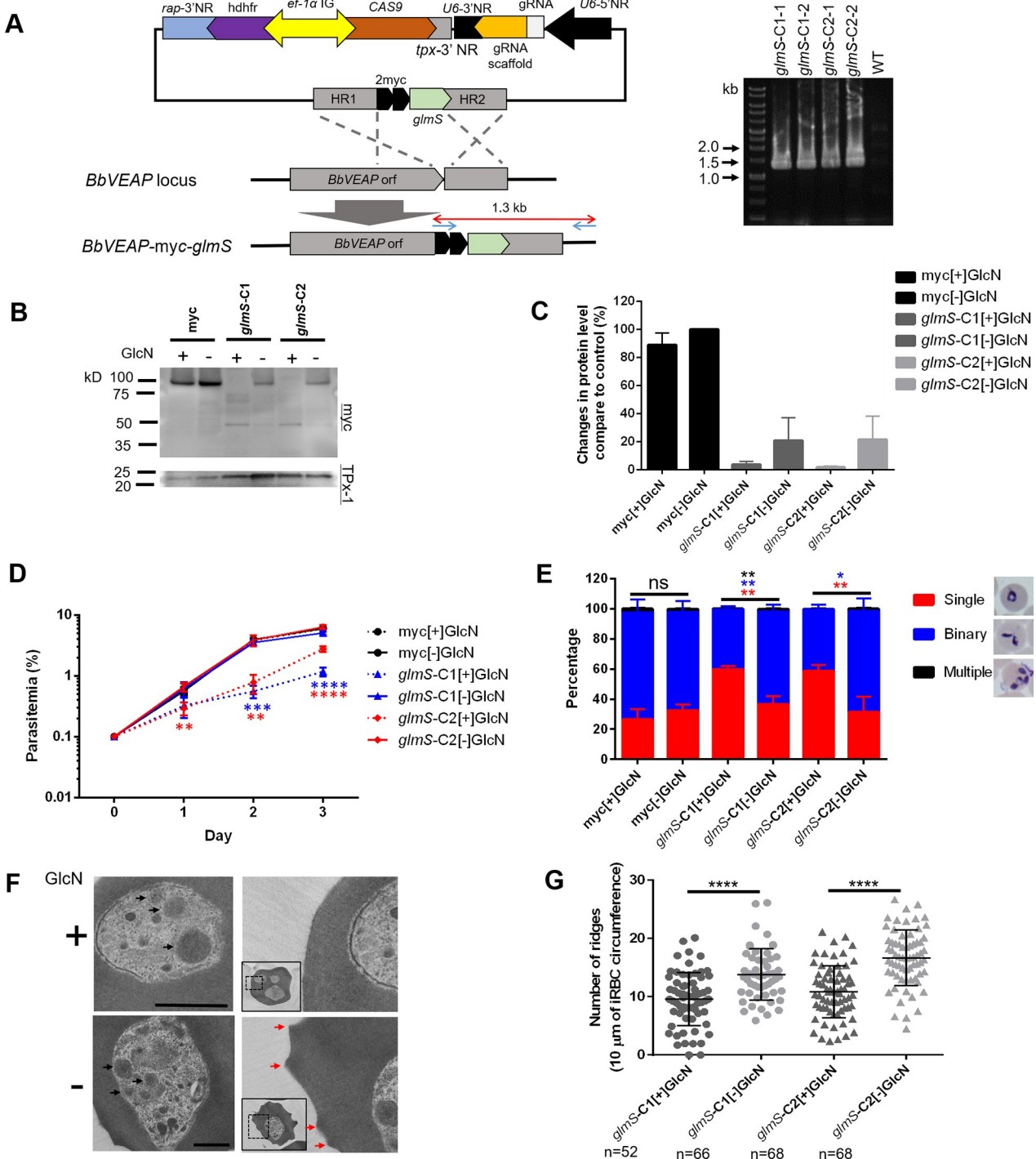

**Fig 5. Induced knockdown of BbVEAP using the *glmS* riboswitch system decreases growth rate and ridge numbers. (A)** Schematic of CRISPR/ Cas9 plasmid to insert myc-*glmS* sequences within the *BbVEAP* gene locus and agarose gel electrophoresis of diagnostic PCR to confirm integration of myc-*glmS* sequence at the 3' end of the *BbVEAP* ORF. *rap*-3'NR, *rhoptry associated protein* 3' noncoding region; *hdhfr* orf, *human dihydrofolate reductase* ORF; *ef-1a* IG, *elongation factor-1α* intergenic region; *tpx*-3'NR, *thioredoxine peroxidase-1* 3' noncoding region; U6-3'NR, *B. bovis U6 spliceosomal RNA* 3' noncoding region; gRNA, guide RNA; and HR, homologous region. *glmS*-C1 and *glmS*-C2 indicate transgenic lines independently generated and following "-1" and "-2" indicate 2 independent clones. **(B)** Western blot analysis of 2 independently generated BbVEAP-myc-*glmS* clones and a control BbVEAP-myc parasite line expressing myc-tagged BbVEAP without the *glmS* element from the endogenous gene locus in the presence or absence of glucosamine (GlcN). Anti-TPx-1 antibody was used to detect TPx-1 protein as a loading control. The image is representative of three independent experiments done within an approximately one-week interval. **(C)** Densitometry of BbVEAP protein levels in all conditions measured relative to the control parasite (GlcN-untreated BbVEAP-myc line) incubated in the presence or absence of GlcN. **(D)** Growth of BbVEAP-myc-*glmS* and control BbVEAP-myc lines in the presence or absence of GlcN. Initial parasitemia was 0.1% and parasitemia was

monitored for 3 days with daily culture medium replacement. The data are shown as mean ± S.D. for three independent experiments performed with a one-week interval. (**, $P < 0.01$; ***, $P < 0.001$; ****, $P < 0.0001$; determined by multiple $t$ test). **(E)** Proportion of ring, binary, and multiple stages in different parasite lines in the presence or absence of GlcN on day 3 post GlcN introduction. The data are shown as mean ± S.D. (*, $P < 0.05$; **, $P < 0.01$; ns, not significant, $P \geq 0.05$; determined by multiple $t$ test). **(F)** Transmission electron microscopy images of BbVEAP-myc-*glmS* in the presence (+) or absence (-) of GlcN. Black arrows indicate spherical bodies and red arrow shows ridges. Scale bar = 0.5 μm. **(G)** Quantification of ridge numbers on the surface of iRBCs of BbVEAP-myc-*glmS* parasites in the presence or absence of GlcN at day 3 post GlcN introduction. Ridge numbers per 10 μm of iRBC circumference were quantified only in mature stage iRBCs (binary form) (****, $P < 0.0001$; determined by Mann-Whitney $U$ test).

VESA1, SBPs, and SmORF, thus validating the study methodology. Of the novel exported protein candidates, 2 Bbmtm proteins belong to a multigene family; and a protein we have termed BbVEAP which is largely conserved within piroplasma. It is noted that our method resulted in the biotinylation and identification of merozoite surface and abundant proteins (S2B Fig), possibly due to biotin entry into or lysis of RBCs and parasites.

The Bbmtm genes are expanded to 44 gene copies in the *B. bovis* genome, the same number as the expansion of *smorf* genes. Like *smorfs*, *mtms* are typically located within gene neighborhoods, often telomeric, containing *ves1* multigene family members. VESA1, SmORFs, and Bbmtms are exported proteins and close association of their gene loci in the genome of *B. bovis* may suggest a common epigenetic control of expression. Although the *mtm* gene family is unique to *B. bovis*, proteins with a similar multi-transmembrane structure such as MFS and TPR are expanded in other piroplasms, indicating a conservation of lineage-specific expansion of multi-transmembrane proteins. Unlike *Plasmodium* and *T. gondii*, which maintain a PV membrane (PVM) through their developmental cycle in the host cell, the PVM ruptures within minutes of invasion by *Babesia*, like a related piroplasm, *Theileria* [39,40]. Therefore, these parasites are in direct contact with the host cell cytoplasm and this may allow the parasite to export proteins possessing multiple TM domains to the host cell. To our knowledge this is distinct from *Plasmodium*, for which no known multi-transmembrane proteins are exported across the PVM. The *Plasmodium* CLAG/RhopH1 proteins that is localized on the iRBC membrane possess two transmembrane domains, although lacking a 'classical'

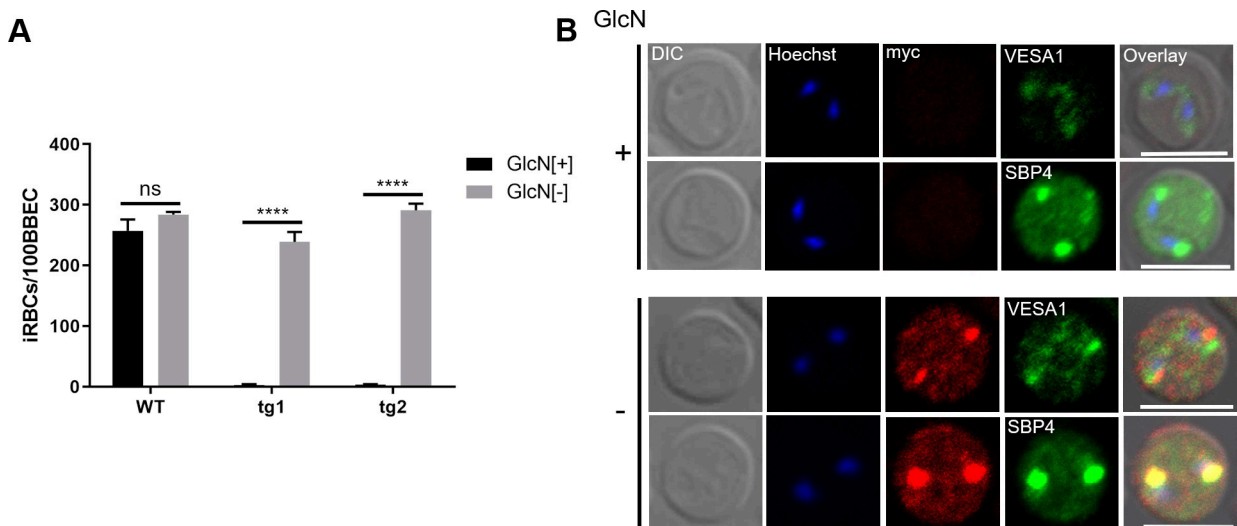

**Fig 6. BbVEAP knockdown abrogates binding of iRBCs to endothelial cells. (A)** Cytoadhesion assay of BbVEAP-myc-*glmS* and WT parasites in the presence (+) or absence (-) of GlcN. All data are expressed as mean ± SEM of triplicate assay (****, $P < 0.0001$; determined by paired Student's $t$ test). **(B)** Indirect immunofluorescence microscopy test of BbVEAP-myc-*glmS* parasite in the presence (+) or absence (-) of GlcN (α-myc, red; α-VESA1 and α-SBP4, green). The parasite nuclei were stained with Hoechst 33342 (Hoechst, blue). Scale bar = 5 μm.

transporter or channel multi-TM structure, and are involved in the permeability pathway across the RBC membrane, also termed PSAC; but they are introduced via merozoite rhoptry secretion during the RBC invasion [32,41,42].

The function of Bbmtm, MFS, or TPR in piroplasms is not known but their structure suggests a transporter or channel activity. It was shown that *B. divergens*, a zoonotic *Babesia* species that infects cattle and immunocompromised humans, increases RBC permeability to various organic solutes, indicating the existence of parasite produced channels or transporters on the surface of iRBC [33]. Similarly, we have seen increased permeability of RBC to sorbitol when they were infected with *B. bovis*, which was reduced in BS-resistant lines. We found that only either *Bb60* or *Bb10* was downregulated in each of two BS-resistant lines among all *mtm* gene family members and revertants recovered the transcription level of respective *mtm*, indicating a link between expression of these *mtms* and BS resistance. These observations support the idea that Bbmtm acts as a channel or transporter, and if this is the case then Bb60-mtm or Bb10-mtm can transport BS across the RBC membrane. Although *Bb60* and *Bb10* were downregulated in BS-resistant lines, the expression of other *mtms* was unchanged in comparison to control parasites, suggesting stable epigenetic regulation, or a substrate specificity for each *mtm*. Transcripts of several other genes were also reduced in the BS-resistant parasite lines, which might contribute to this resistance, and needs future evaluation.

In this study we identified BbVEAP and found that this protein is involved in ridge formation, VESA1 expression on iRBC, and binding of iRBCs to endothelial cells, which establishes BbVEAP as a novel virulence factor of *B. bovis*. Although distribution of VESA1 in the iRBC was affected by BbVEAP knockdown, the mechanism behind this observation is unclear. Because we could not co-precipitate these two proteins by immunoprecipitation experiments (S8B Fig), they may not directly interact. Recently it was shown that upregulation of SBP2 truncated copy 11 in *B. bovis* reduced binding of iRBCs to endothelial cells and its virulence in cattle [43,44]. However, it is unclear how SBP2 truncated copy 11 affects the export or surface expression of VESA1 or other putative cytoadhesion ligands. A knobless line of *P. falciparum* has been described, and though the surface expression of PfEMP1, the functional homolog of VESA1 in *P. falciparum*, is reduced in these parasites they remained able to bind to endothelial cells [45,46]. Ridgeless *B. bovis* has not been reported and VESA1 is the only known ligand for cytoadhesion of *B. bovis* iRBCs [4]. Depletion of BbVEAP abrogated cytoadhesion indicating its role in the export or distribution of VESA1 and/or other putative ligands to the iRBC surface. Additionally, BbVEAP has a single ortholog in all piroplasmida that includes parasite species lacking ridge-like structures on iRBCs, suggesting a piroplasm-conserved function other than cytoadhesion. Considering the essentiality of this gene and conservation across piroplasmida, including the genera *Theileria*, targeting this protein could be promising for the development of pan piroplasmida therapeutics.

Spherical bodies are large secretory organelles within the apical end of *Babesia* spp. [47]. The products of two multigene family in *B. bovis*, SBP2 and SmORFs, are localized in spherical bodies [9,15]. In this study we found two novel proteins released from spherical bodies to iRBC, Bbmtm and BbVEAP. It was proposed that the PLM in SBP2 and SmORFs works as a retention signal in spherical bodies [15]. Bbmtm and BbVEAP also have PLM, suggesting the importance of PLM for the protein export to iRBC via spherical bodies in *Babesia* parasites.

Surface exposed proteins are targets of immunity and thus are vaccine targets, such as PfEMP1-VAR2CSA which is expressed on the surface of *P. falciparum* iRBCs and is responsible for pathogenesis of pregnancy-associated malaria [48]. VAR2CSA-based vaccines are currently in clinical trials [49]. The *Plasmodium* new permeability pathway, PSAC, is produced by *Plasmodium* and is inserted into the iRBC membrane for nutrient acquisition. Since nutrient acquisition is essential for malaria parasite survival, PSAC is considered a promising target for

the development of antimalarials [50]. Targeting channels or transporters on the *Babesia*-iRBC membrane may lead to the development of pan anti-babesiosis drugs. Further characterization of our proteome data set could pave the way for identification of novel vaccine and drug targets and further development of new control strategies for bovine babesiosis caused by *B. bovis*.

## Methods

### Parasite culture and transfection

*B. bovis* Texas strain was obtained from Washington State University and kept in continuous culture using a microaerophilic stationary-phase culture system composed of purified bovine RBCs at 10% hematocrit and GIT medium (Wako Pure Chemical Industries, Japan).

The transfection of *B. bovis* was done as described [51,52]. Briefly, 100 µL of *B. bovis*-iRBCs were mixed with 10 µg of plasmid constructs in 100 µL of Amaxa Nucleofector human T-cell solution. Transfection was done using a Nucleofector device, program v-024 (Amaxa Biosystems, Germany). Ten nanomolar WR99210 was added one day after the transfection to select a transgenic parasite population.

### Enrichment of *B. bovis*-iRBCs

Enrichment of *B. bovis*-iRBCs was done using a Histodenz solution. The solution was prepared by dissolving 27.6 g of Histodenz (Sigma-Aldrich) in 100 mL of Tris-buffered solution (5 mM Tris-HCl, 3 mM KCl, and 0.3 mM $Na_2$-EDTA, pH 7.5). The iRBCs were layered on the surface of 80% Histodenz solution in GIT medium and centrifuged for 30 min at 2500 x g with a swinging bucket rotor. The iRBCs at the bottom of the tube were used for biotinylation assays.

### Biotinylation and protein extraction

Biotinylation of iRBC surface proteins was done as described [25,37]. In the first attempt, proteins were serially extracted using a hypotonic solution (20X diluted PBS) and BugBuster protein extraction reagent (Novagen) containing the endonuclease benzonase. MS showed high bovine hemoglobin contamination. Thus, in the second and third attempts biotinylated iRBCs were initially treated with 0.2% (w/v) saponin on ice for 15 min to remove hemoglobin. Protein extraction was done serially first by resuspending the parasite pellets in BugBuster protein extraction reagent containing the endonuclease benzonase. Following centrifugation and removing extracted proteins, the remaining parasite pellet was lysed with a solution containing 150 mM NaCl, 5 mM EDTA, 50 mM Tris pH8.0, 1.0% Triton-X 100 (w/v), and protease inhibitor cocktail (Complete Mini, Roche) at 4˚C for 1 h. The protein extract was incubated with Dynabeads MyOne Streptavidin C1 (Invitrogen) with rotation at 4˚C for 1 h. The beads were washed 5 times with wash buffer containing 0.1% SDS, 400 mM urea, 150 mM NaCl, and 50 mM Tris-HCl (pH 8.0) to remove unbound proteins. Bound proteins were eluted from the beads by boiling for 5 min in 1x Sample Buffer (25 mM Tris (pH 6.8), 2.5% w/v SDS, 2.5% v/v glycerol, 0.08% w/v bromophenol blue, and 5% beta-mercaptoethanol). Proteins were assessed for biotinylation via Western blotting using horseradish peroxidase (HRP)-conjugated streptavidin (1:40,000, Invitrogen).

### Liquid chromatography-tandem mass spectrometry (LC-MS/MS)

The purified biotinylated proteins were subjected to LC-MS/MS as described [53]. The samples were briefly electrophoresed on SDS-polyacrylamide gel electrophoresis. The gel containing proteins were excised, fixed with acetic acid/methanol solution, and subjected to LC–MS/

MS analysis at the W. M. Keck Biomedical Mass Spectrometry Laboratory, University of Virginia, USA. The data analysis was performed by database searching using the Sequest search algorithm against the *Bos taurus* and *B. bovis* reference strain in UniProt and SwissProt. Filtering and extraction of data was performed using Scaffold version 4.8.9 (Proteome Software Inc.). Protein identifications were accepted if they could be established at greater than 90% probability and contained at least 1 identified peptide. Quantitative value (normalized total spectra) was used to show an estimate of protein abundance between biotinylated and non-biotinylated samples. PiroplasmaDB-34 [54] was used for annotation of the target proteins.

## Cytoadhesion assay

Cytoadhesion assays were done as described [55]. Briefly, bovine brain endothelial cells (BBECs; Cell Applications Inc., USA) were seeded in 6 well plates containing cover glasses (Matsunami Glass, Japan). *B. bovis*-iRBCs with 2–5% parasitemia and 1% hematocrit were added to the BBEC culture. The cells were incubated for 90 min with gentle agitation every 15 min. Nonadherent iRBCs were washed away with Hanks-balanced salt solution. Cells on the cover glasses were fixed with methanol and stained with Giemsa's solution and the number of bound iRBCs were counted for 500 BBECs.

## Plasmid construction

The schematic of the plasmid expressing myc-tagged target proteins is shown in S2 Fig. The primers used for plasmid construction are listed in S6 Table. *B. bovis elongation factor-1α* intergenic region-B (*ef-1α* IG-B), ORF of the gene of interest (GOI), and *thioredoxine peroxidase-1* (*tpx-1*) 3' noncoding region (NR) were PCR-amplified from *B. bovis* genomic DNA. A DNA fragment containing *B. bovis actin* 5' NR (*act* 5'NR), *human dihydrofolate reductase* (*hdhfr*), and *B. bovis rhoptry associated protein* 3' NR (*rap* 3'NR) was amplified from a *B. bovis* green fluorescent protein (GFP)-expressing plasmid [51]. *ef-1α* IG-B was cloned into the EcoRI site of pBluescript SK using In-Fusion HD Cloning Kit (Takara Bio Inc., Japan). Subsequently, the ORF of GOI tagged with 2 myc epitopes, *tpx-1* 3'NR, and *act* 5'NR-*hdhfr*-*rap* 3'NR were cloned into SmaI to make the final plasmid for episomal expression of myc-tagged proteins.

The CRISPR/Cas9 system was employed to delete BBOV_III004280 or insert myc and *glmS* sequences into the 3' end of the original locus [56]. Briefly, homologous regions (HRs) were PCR-amplified from *B. bovis* genomic DNA and inserted into the BamHI site of the BbU6-Cas9-hDHFR plasmid. Single guide RNA (sgRNA) was inserted into the AarI site of BbU6-Cas9-hDHFR using T4 DNA ligase (New England Biolabs, USA). Parasites were transfected as described [51,52] and the obtained transgenic parasites were cloned by limiting dilution before analysis. Diagnostic PCR was done using glms-F-IF and 4280-3NR-integR (S6 Table) to confirm insertion of myc-*glmS*.

## Development of BS-resistant *B. bovis* and comparative transcriptomics

*B. bovis* WT parasites were initially cultured with 1 μg/mL blasticidin-S solution (BS; Thermo Fisher Scientific, USA), and the BS concentration was increased stepwise. Two independent resistant lines that propagate under 4 μg/mL BS were produced. To produce revertant parasites, BS-resistant lines were cultured without BS for two months. Comparative transcriptomics between *B. bovis* WT and its BS-resistant derivatives or revertant parasites were performed by RNA-seq. RNA was extracted from parasites using TRIzol reagent (Invitrogen). Libraries were constructed using a TruSeq Stranded mRNA Library Preparation Kit (Illumina, USA), according to the manufacturer's protocol and the products were subjected to

Novaseq6000 (Illumina) with the 150-bp paired-end protocol. Acquired reads were mapped against *B. bovis* T2Bo reference genome obtained from PiroplasmaDB-37 using HISAT2 [57]. Read data normalization and differential expression were obtained using Cufflinks with the default parameters [58]. Complementary DNA was synthesized using SuperScript III Reverse Transcriptase (Invitrogen) with random primers.

## Bioinformatics analysis

SignalP-5.0 and -3.0 were used to predict putative signal peptides [59]. TMHMM-2.0 was used to predict TM domains [60]. GPI anchors were predicted using PredGPI [61]. To identify multigene families encoding multiple TM domains, whole genome sets of protein sequences for *B. bovis*, *B. bigemina*, *B. ovata*, *B. microti*, *Theileria annulata*, *T. parva*, *T. orientalis*, *P. falciparum*, and *T. gondii* were evaluated using TMHMM-2.0 and proteins were selected having equal to or more than eight TM domains and at least one TM domain average per 100 amino acids. Mutual homology among the selected proteins were identified by BLASTP [62] and proteins with more than a 200 bit-score and by eye were regarded as homologs. The overall relationships were visualized with Gephi [63] using a Fruchterman–Reingold layout.

## SDS-PAGE and Western blotting

Parasite proteins were extracted using 1.0% Triton-X 100 (w/v) in PBS and protease inhibitor cocktail (Complete Mini, Roche) at 4˚C for 1 h. The protein fractions were separated by electrophoresis and transferred to polyvinylidene difluoride membranes (Millipore, USA). The membranes were probed with mouse anti-myc monoclonal antibody (1:500; 9B11, Cell Signaling Technology, USA), rabbit anti-TPx-1 polyclonal antibody (1:250 [52]), rabbit anti-VESA1 polyclonal antibody (1:100), or rabbit anti-SBP4 polyclonal antibody (1:1000 [13]) at 4˚C overnight. Washing was done in PBS supplemented with 0.05% Tween-20 (TPBS). Secondary probe of membranes was done with HRP-conjugated goat anti-mouse or rabbit IgG (1:8,000; Promega, USA). Protein bands were visualized using ECL Select Western Blotting Detection Reagent (GE healthcare) and detected by a chemiluminescence detection system (LAS-4000 mini; Fujifilm, Japan).

Immunoprecipitations were performed as described [64]. Parasite pellets were prepared by saponin treatment of BbVEAP-myc tagged parasites and cross-linked with 1 mM DSP (Sigma). Proteins were extracted with 1% Triton X-100 (w/v) in PBS (containing 1 mM EDTA, 10% glycerol, and protease inhibitors) at 4˚C for 1 h. The extracted proteins were incubated with anti-myc mouse mAb (9B11, Cell Signaling) at room temperature for 4 h with gentle rotation. The solution was then mixed with protein G Sepharose 4 fast flow (GE Healthcare) and incubated with rotation at 4˚C overnight. The mixture was centrifuged and the beads were washed with 0.5% Triton X-100 (w/v) in PBS (containing 1 mM EDTA, 10% glycerol, and protease inhibitors). To elute proteins, the beads were mixed with 2.4 μg/μL of c-myc peptide (Thermo Fisher Scientific) and incubated at 4˚C for 12 h. The beads were centrifuged and the supernatant was collected as an immunoprecipitated (IP) fraction.

## Indirect immunofluorescence antibody test (IFAT)

IFAT was done on thin blood smears from cultured parasites that had been air-dried and fixed in a 1:1 acetone:methanol mixture at −20˚C for 5 min [65]. Smears were immunostained with mouse anti-myc monoclonal antibody (9B11) at 1:500 dilution in TPBS and incubated at 37˚C for 60 min. Double immunostaining of smears was done with rabbit anti-VESA1α (antisera against peptide YNQVVHYIRALFYQLYFLRK; Medical & Biological Laboratories Co., ltd, Japan) at 1:50 dilution, or rabbit anti-SBP4 at 1:1000 dilution. The smears were incubated with

Alexa fluor 488- or 594-conjugated goat anti-mouse or Alexa fluor 488-conjugated goat anti-rabbit IgG antibody (1:500; Invitrogen) at 37˚C for 30 min. For staining the nuclei, the smears were incubated with 1 μg/mL Hoechst 33342 solution at 37˚C for 20 min. The smears were examined using a confocal laser-scanning microscope (A1R; Nikon, Japan).

### Electron microscopy

For preparation of transmission electron microcopy (TEM) samples, iRBCs were fixed with 2% glutaraldehyde (Nacalai Tesque, Japan) in 0.1 M sodium cacodylate buffer at 4˚C for 60 min. The samples were rinsed, and then post-fixed with 1% $OsO_4$ (Nacalai Tesque) at 4˚C for 60 min. The samples were washed, dehydrated in a graded series of ethanol and acetone, and embedded in Quetol 651 epoxy resin (Nisshin EM, Japan). Ultra-thin sections were stained and examined at 80 kV under a transmission electron microscope (JEM-1230; JEOL, Japan).

Immunoelectron microcopy samples were fixed with 4% paraformaldehyde and 0.1% glutaraldehyde in 0.1 M phosphate buffer (PB) on ice for 15 min. Cells were dehydrated with 30%, 50%, 70%, and 95% ethanol each time at 4˚C for 5 min, and embedded in LR White resin (London Resin Company, UK). Thin sections were blocked with 5% non-fat milk (Becton, Dickinson and Company, USA) and 0.0001% Tween 20 (Wako, Japan) in PBS (PBS-MT) at 37˚C for 30 min and incubated with rabbit anti-myc polyclonal antibody (1:100; ab9106, Abcam, UK) or control normal rabbit IgG at 4˚C overnight. After washing the sections were incubated with goat anti-rabbit IgG conjugated to 15 nm gold particles (1:20; EY Laboratories Inc., USA), and fixed with 0.5% $OsO_4$ at room temperature for 5 min. The sections were stained and observed by TEM.

For scanning electron microscopy (SEM), the iRBCs were enriched by density gradient separation using a Percoll and sorbitol solution [33]. Samples were fixed with 1.2% glutaraldehyde in 0.1 M PB at room temperature for 20 min. They were then rinsed and post-fixed with 1% $OsO_4$ at room temperature for 10 min. After dehydration in graded series of ethanol, the samples were immersed in t-butyl alcohol and placed at -20˚C overnight, and then freeze-dried. The samples were sputter-coated with gold and palladium and imaged on a scanning electron microscope (JSM-840; JEOL).

### Statistical analyses

The parasitemia and proportion of parasite stages were plotted using Prism 6 (GraphPad Software, USA) and evaluated using multiple *t*-test. The ridge numbers on the surface of iRBCs were compared using Mann-Whitney *U* test. The difference in number of cytoadhered iRBCs per 500 BBECs was evaluated using a two-tailed paired Student's *t*-test. The values were considered significantly different if *P*-value was below 0.05.

### Supporting information

**S1 Fig. Panning of *B. bovis* and live fluorescence microscopy of biotinylated iRBCs. (A)** Selection of cytoadherent *B. bovis* to BBECs. The number of bound iRBCs per 100 BBECs were counted. **(B)** Live fluorescence microscopy of biotinylated iRBCs reacted with streptavidin-conjugated Alexa Fluor 488 (green). The parasite nuclei were stained with Hoechst 33342 (Hoechst, blue). No released merozoites were seen. Scale bar = 10 μm.
(TIF)

**S2 Fig. Schematic of a plasmid expressing myc-tagged candidate proteins and fluorescence microscopy images. (A)** Schematic of a plasmid expressing myc-tagged candidate proteins. *ef-1αIG-B*, elongation factor-1α intergenic region B; *GOI* orf, *gene of interest* ORF; *tpx*-3'NR,

*thioredoxine peroxidase-1* 3' noncoding region; *hdhfr* orf, *human dihydrofolate reductase* ORF; *rap*-3'NR, *rhoptry associated protein* 3' noncoding region. **(B)** Indirect immunofluorescence antibody test of transgenic *B. bovis* expressing myc-tagged target proteins stained with anti-myc (green). The parasite nuclei were stained with Hoechst 33342 (Hoechst, blue). Scale bar = 5 μm.
(TIF)

**S3 Fig. Quantification of gold particles in immunoelectron micrographs.** The number of gold particles were quantified in *B. bovis* with Bb60-mtm and BbVEAP tagged with myc epitopes. Gold particle numbers were counted in spherical bodies and iRBC surface only in parasites with clear spherical bodies in electron micrograph sections (****, $P < 0.0001$; determined by Mann-Whitney $U$ test).
(TIF)

**S4 Fig. Quantitative reverse transcription PCR of *Bb60* and *Bb10*.** Relative transcript levels of *Bb60* and *Bb10* in WT, BS-resistant lines and BS-sensitive revertants. Transcript levels are normalized against *methionyl-tRNA synthetase* (Gene ID: BBOV_I001970).
(TIF)

**S5 Fig. Overexpression of Bb60-mtm in WT line did not affect the BS resistance.** Growth inhibition curves of two Bb60-overexpressing parasite lines in the presence of different concentrations of BS (μg/mL). Bb60-mtm is episomally overexpressed in WT parasites under 10 or 100 nM WR99210 in Bb60-10 or Bb60-100 lines, respectively. All data are expressed as mean ± SEM.
(TIF)

**S6 Fig. Growth of BbVEAP-myc-*glmS* parasite under increasing concentrations of GlcN.** Growth of BbVEAP-myc-*glmS* and control parasites (BbVEAP-myc) in the absence or presence of 1, 2.5, 5, and 10 mM GlcN. Initial parasitemia was 0.05% and parasitemia was monitored for 3 days with daily culture medium replacement. The data are shown as mean ± S.D. from technical triplicates.
(TIF)

**S7 Fig. Scanning electron microscopy of BbVEAP-myc-*glmS* parasite in the absence or presence of GlcN.** Scanning electron microscopy showing ridges on the surface of RBC infected with BbVEAP-myc-*glmS* parasites or a control parasite following 3 days exposure to GlcN. Scale bar = 1 μm.
(TIF)

**S8 Fig. Western blot analysis of BbVEAP-myc-*glmS* and WT parasites and immunoprecipitation of BbVEAP-myc parasite. (A)** Western blot analysis of two clones (tg1, tg2) of BbVEAP-myc-*glmS* and WT parasites in the presence or absence of GlcN. TPx-1 detected with anti-TPx-1 antibody was used as a loading control. **(B)** Immunoprecipitation with anti-myc for BbVEAP-myc parasite.
(TIF)

**S1 Table. List of identified proteins by mass spectrometry.**
(XLSX)

**S2 Table. List of predicted secretory proteins.**
(XLSX)

**S3 Table. List of putative exported proteins confirmed for their localization.**
(XLSX)

**S4 Table. List of *mtm* in the *B. bovis* genome.**
(XLSX)

**S5 Table. Genes up- or downregulated in BS-resistant parasites.**
(XLSX)

**S6 Table. List of primers used in this study.**
(XLSX)

## Acknowledgments

We thank Dr. Ikuo Igarashi, Obihiro University of Agriculture and Veterinary Medicine, Japan for supplying anti-SBP4. This work was conducted at the Joint Usage/Research Center on Tropical Disease, Institute of Tropical Medicine, Nagasaki University, Nagasaki, Japan, and at the National Research Center for Protozoan Diseases, Obihiro University of Agriculture and Veterinary Medicine, Hokkaido, Japan.

## Author Contributions

**Conceptualization:** Hassan Hakimi, Masahito Asada.

**Formal analysis:** Hassan Hakimi, Osamu Kaneko, Masahito Asada.

**Funding acquisition:** Hassan Hakimi, Shin-ichiro Kawazu, Osamu Kaneko, Masahito Asada.

**Investigation:** Hassan Hakimi, Thomas J. Templeton, Miako Sakaguchi.

**Methodology:** Hassan Hakimi, Junya Yamagishi, Shinya Miyazaki, Kazuhide Yahata, Takayuki Uchihashi.

**Project administration:** Masahito Asada.

**Resources:** Hassan Hakimi, Shin-ichiro Kawazu, Osamu Kaneko, Masahito Asada.

**Supervision:** Masahito Asada.

**Validation:** Hassan Hakimi, Masahito Asada.

**Visualization:** Hassan Hakimi, Thomas J. Templeton, Miako Sakaguchi, Junya Yamagishi, Masahito Asada.

**Writing – original draft:** Hassan Hakimi, Thomas J. Templeton.

**Writing – review & editing:** Hassan Hakimi, Thomas J. Templeton, Miako Sakaguchi, Junya Yamagishi, Shinya Miyazaki, Kazuhide Yahata, Takayuki Uchihashi, Shin-ichiro Kawazu, Osamu Kaneko, Masahito Asada.

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
