## [Decision Letter · Decision Letter 0]

28 Mar 2020

Dear Dr. Asada,

Thank you very much for submitting your manuscript "Novel Babesia bovis exported proteins that modify properties of infected red blood cells" for consideration at PLOS Pathogens. As with all papers reviewed by the journal, your manuscript was reviewed by members of the editorial board and by several independent reviewers. In light of the reviews (below this email), we would like to invite the resubmission of a significantly-revised version that takes into account the reviewers' comments.

Your submission was evaluated by three independent reviewers and found to contain novel information of potential significance to our understanding of Babesia spp. parasites. However, significant concerns were raised, including the starting point of surface biotinylation. Of particular significance are concerns regarding the veracity of some conclusions that were not felt to be adequately supported by the data provided. For example, stronger rationale is needed justifying the focus on Bb60 and Bb10 when numerous genes underwent significant transcriptional changes. It was felt that conclusions about function of Bbmtm in nutrient acquisition, and of BbVEAP in cytoadhesion, were of potential significance but inadequately supported. If you choose to respond by minimizing your claims most of these issues may be dealt with by rewriting. If you choose to gain more experimental evidence to support your claims this could amount to a significant amount of effort. Your revision should address the specific points raised by each reviewer.

We cannot make any decision about publication until we have seen the revised manuscript and your response to the reviewers' comments. Your revised manuscript is also likely to be sent to reviewers for further evaluation.

Sincerely,

David R. Allred, Ph.D.

Guest Editor

PLOS Pathogens

Kirk Deitsch

Section Editor

PLOS Pathogens

Kasturi Haldar

Editor-in-Chief

PLOS Pathogens

orcid.org/0000-0001-5065-158X

Michael Malim

Editor-in-Chief

PLOS Pathogens

orcid.org/0000-0002-7699-2064

Reviewer's Responses to Questions

**Part I - Summary**

Reviewer #1: (No Response)

Reviewer #2: Hakimi et al. show in their manuscript entitled 'Novel Babesia bovis exported proteins that modify properties of infected red blood cells' by a combination of surface biotinylation/LcMSMS and follow-up studies involving inducible knockouts the identification of a new group of proteins, presumed transporters in the RBC membrane and another, independent protein essential for host cell modifications resulting in cytoadherence.

A large amount of work clearly went into this manuscript. The manuscript is well written, the experiments well conducted and the results are interesting, not only to people in the field of Babesia biology, but also for people interested in host cell modifications.

However, the manuscript could improve in the following ways:

Firstly, the authors do not make it clear that the surface biotinylation approach resulted, like it did previously in other species, in leakiness of the RBC and/or stickiness of other, non-surface proteins, requiring drastical filtering of the 255 B. bovis hits, to come to a list of 38 proteins. This list still includes a number of known merozoite surface proteins, not only parasite proteins translocated to the RBC surface. Indeed, the authors show in S2 Fig, that 7 of the 10 uncharacterised proteins from that list, are localised to either apical organelles or the merozoite surface. Only 3 show the targeted export into the host cell the authors were looking for in their biotinylation experiment. Therefore, the authors should make it clear that the surface labelling did in fact label most abundant proteins, both within the RBC and probably also within the parasite and therefore the list of 38 resembles a list of proteins for which masspec evidence is available in blood stages and which have additionally one of the following criteria: signal sequence, TM domain or ER retention signal.

Secondly, the flow of the manuscript is lacking. The authors could improve this by highlighting that once they generated the list of expressed proteins in B.bovis which have one of the 3 criteria stated above, that they focussed on 2 of the genes from their list of 38 which showed export into the RBC. One of these is a member of the novel multigene family of potential transporters, which is a very interesting finding. The authors state that in B. bovis this list has 44 members but fail to include this list in the supplementary materials. In order to demonstrate that at least some of these MTMs are not only localized to the RBC membrane but also function as transporters the authors conduct RNA-seq analysis on BSD resistant B. bovis lines, identifying reduced expression of Bb60 and Bb10. This is comparable to studies in Plasmodium falciparum where the same has been described. Presumably Bb10 is a member of the MTM family which was not identified my the masspec approach? Also, presumably the authors tried to generate knockouts of Bb60 without detecting a growth or adhesion phenotype? Did the authors follow up on the function of Bb11920, but failed to find a phenotype and hence dropped the characterisation?

Following on from two MTM family members the authors then started to characterise VEAP, again without a clear link why the reader has to jump to this molecule next. The authors should clearly state that the chosen proteins for characterisation are the ones which they demonstrated earlier of being exported.

To characterise the function of VEAP, the authors use very elegantly the GlmS system to inducibly down-regulate the transcript levels upon glucosamine addition. Unclear is however why even before GlcN addition the protein levels are reduced by 80% compared to wt. Could the authors please address this and speculate? 20% of protein level of VEAP clearly was sufficient to perform a wildtype-like function, and the further reduction by GlcN addition resulted in the recorded phenotype which is striking and an important contribution to RBC surface modification mechanisms in Babesia.

Reviewer #3: Review for Hakimi et al: this manuscript describes the surface biotinylation of Babesia bovis infected erythrocytes and the validation of a selection of the identified hits. It reveals 3 new exported proteins from B. bovis. Two belong to a group of multi transmembrane host cell surface proteins encoded by a gene family termed Bbmtm and one that was termed VEAP, a protein without transmembrane domain that localizes to the host cell cytoplasm. The authors go on to present functional data for both, the Bbmtm and VEAP. This includes the analysis of a conditional VEAP knock down that indicates a role of this protein in cytoadhesion of B. bovis infected erythrocytes and a role in formation of host cell ridges. The authors also postulate that Bbmtm are involved in the nutrient permeability of the membrane of infected erythrocytes.

Overall this is a very interesting paper that identifies novel exported proteins that play (or in the case of Bbmtm products may play) important roles for B. bovis. However, there are a number of concerns that need to be considered. While the identification of Bbmtm gene products as novel host cell membrane proteins is convincing and the comparative analysis of this family in Babesia and related parasites is important, the functional insight into these proteins is very preliminary. These findings should either be substantiated (overexpression of Bb60 or Bb10 in the blasticidine resistant lines would be an easy experiment to do, see major points) or the conclusions on the functon of these proteins needs to be toned down or even removed. The data on VEAP is stronger but the influence on VESA1 is not very broadly documented and should also be substantiated. If these points can be thoroughly addressed, this will result in a very nice study that will be of high interest to the field.

**Part II – Major Issues: Key Experiments Required for Acceptance**

Reviewer #1: (No Response)

Reviewer #2: (No Response)

Reviewer #3: Major points:

1. The data on a role of Bbmtm gene products in nutrient permeability of the host cell is not very conclusive. The authors raise parasites resistant to blasticidine and show that this is due to a reduced permeability of the membrane of infected erythrocytes. Transcriptomics with these parasites showed that in the two blasticidine resistant clones either Bb60 or Bb10 (both Bbmtm members) were downregulated - but among quite a large list of other genes, some of which were much more downregulated. Causality between downregulation of a Bbmtm and the phenotype is therefore not established. For instance VESA1 is downregulated much more in both cases, but likely is not the reason for the observed phenotype. The same is the case for many other downregulated genes. Bb10 and Bb60 are also not the most dramatically regulated genes.

Figure S3 shows that Bb10 is expressed more in Res1 than in the corresponding reverted parasites and this difference seems to be in a similar fold level to the upregulation of Bb10 in the Reverted 2 vs Res2 line, indicating that a certain level of randomness is involved.

To establish the role of the Bbmtm in the nutrient permeability of the infected erythrocyte more experiments are needed. Instead of overexpressing Bb60 in wild type parasites, overexpression of the downregulated Bb60 or Bb10 should be carried out in the R1 and R2 lines. Only if this reverts the phenotype, a function of these proteins in the channel activity can be postulated. This experiment is essential to draw the conclusion that these proteins are involved in this activity, otherwise these statements need to be removed from the paper (e.g. such as line 385...: 'This resistance was linked to the expression of one mtm (Bb10 and Bb60 in each line) indicating that mtm is responsible for BSD uptake and may form a channel or transporter on the surface of iRBC...').

Some other remarks related to this point:

- If Bbmtm indeed turn out to play a role in the nutrient channel function of the host cell, I wonder why only very few Babesia species have an expanded family of these proteins if they have this function.

- The fact that (line 364...): 'VESA1, SmORFs, and Bbmtms are exported proteins and close association of their gene loci in the genome of B. bovis may suggest a common epigenetic control of expression.' might also be used for a reversed argument and could explain why Bbmtms were downregulated, even though they may not have a function in this activity (if another neighbouring gene is involved in this activity and all other genes in the vicinity are epigenitically co-regulated).

2. The functional data on BbVEAP is more direct and more broadly founded than that on Bbmtm. The conditional knock down of this protein convincingly shows a reduced presence of ridges and reduced cytoadherence. However, the validation of the knock down is only shown for the non-cytoadherent parasites. The full validation should also be shown for the cytoadherent parasites (this could for instance be provided as a supplementary figure). More information should also be given about this cytoadherent line (origin; characteristics; was it used before?). Furthermore the effect on VESA1 needs to be more thoroughly analysed. In Fig. 6b, in the GlcN+ parasites the VESA1 signal simply seems very faint. Is there a real difference in the location? Can the change in distribution (or expression levels?) of VESA1 be substantiated and quantified (IFA and Westernblot)? Is there a change in the distribution of VESA1 on the erythrocyte surface in EM?

How do the authors explain that VEAP appears to be soluble in the host cell but was detected by the surface biotinylation? What does this mean for the other hits found in their mass spectrometry analysis? How can VEAP1 influence VESA1 and ridge formation in this location?

**Part III – Minor Issues: Editorial and Data Presentation Modifications**

Reviewer #1: (No Response)

Reviewer #2: - Include number of unique peptides identified per protein in table S1

- Line 126 states that a complete list of all identified proteins is given, it should read all identified B. bovis proteins as the host proteins are not listed.

- Include a list of all 44 B.bovis MTM genes

- It would be desirable to verify that other MTM proteins identified by masspec in this study also localize to the RBC membrane. Why choose Bb60 and Bb11920 out of the list of 7 mtm family proteins?

Reviewer #3: Minor points:

- The authors report that 32 of the proteins identified by the surface biotinylation possessed a

PEXEL-like export motif (RxLx or RxxLx). Please clarify where in the protein these motifs were found. In malaria parasites these motifs are only functional if in a specific range downstream of a signal peptide but this is different in T. gondii. What were the criteria for calling of such a motifs in this study? I could not find the information in the M&M how this was done.

- Twelve proteins, including two positive controls, were selected for validation (line 168... and Figure S2). Can the authors explain in the text how exactly these proteins were selected?

- Please provide statistics for the gold particle counts shown in Fig. 2c (how cells were selected; number of cells inspected; location of gold etc)

- Why do Bbmtm A and B cluster so far apart in 3c? Does this mean they are two independent families? What is the level of sequence homology?

- Figure 5a PCR gel. I assume these are cloned parasites. If not, it should be shown that there are no parasites without the correct genomic insertion using a PCR to detect the unmodified locus (to exclude the presence of parasites with incorrect insertions). What is tg1-1 and tg1-2 etc in this figure part?

- there is a 'data not shown' in the discussion, please either provide the data or remove

- indicate at least fixative used for the IFA procedure (line 557 'air-dried and fixed as described [62]'.), so the reader does not need to consult other publicatons to know the type of IFA that was used.

- please provide the source of rabbit anti-SBP4 antibody (or provide validation if newly generated)

- is this a commercial rabbit anti-VESA1α antibody by MBL? If it is a custom made antibody, again validation of these antibodies needs to be provided

- How were the CRISPR lines obtained? Were parasites cloned? If yes, how?

- exact information how often experiments were done and if technical or biological replicas are shown is often missing. Just as an example, the legend of figure 6 A says that all data are 'triplicate assays'. Are these independent experiments or technical replicas?

- please remove the statement in line 390-392: if this is true, this manuscript at least does not show it.

PLOS authors have the option to publish the peer review history of their article (what does this mean?). If published, this will include your full peer review and any attached files.

Reviewer #1: No

Reviewer #2: No

Reviewer #3: No
---

## [Decision Letter · Decision Letter 1]

29 Jul 2020

Dear Dr. Asada,

Thank you very much for submitting your manuscript "Novel Babesia bovis exported proteins that modify properties of infected red blood cells" for consideration at PLOS Pathogens. As with all papers reviewed by the journal, your manuscript was reviewed by members of the editorial board and by several independent reviewers. The reviewers appreciated the attention to an important topic. Based on the reviews, we are likely to accept this manuscript for publication, providing that you modify the manuscript according to the review recommendations.

Please note that Reviewer 3 has identified a few remaining points in need of correction before the manuscript could be accepted. Of particular importance are two points: (i) The reviewer makes a significant point that the data in S5 Figure is important to your arguments. Please try either to combine it into one of the main figures, or make it a main figure. (ii) Please respond to the comment regarding correction of labeling in S8 Figure to avoid any confusion. Upon inspection, I observed two issues in S5 Figure: (i) the x-axes are labeled in “log ug/mL”, whereas in the figure legend the concentrations are provided in nM. Please make the units of concentration consistent throughout the manuscript (either system, but preferably in molarity). (ii) In the lower panel of Figure S5B there are 4 curves but only 3 identified samples; please correct this. In addition, Figure S1B shows evidence of significant pixel shift between the DIC and fluorescent images. Please realign the superimposed images for correct alignment. Thank you for your attention to these matters.

Sincerely,

David R. Allred, Ph.D.

Guest Editor

PLOS Pathogens

Kirk Deitsch

Section Editor

PLOS Pathogens

Kasturi Haldar

Editor-in-Chief

PLOS Pathogens

orcid.org/0000-0001-5065-158X

Michael Malim

Editor-in-Chief

PLOS Pathogens

orcid.org/0000-0002-7699-2064

Reviewer Comments (if any, and for reference):

Reviewer's Responses to Questions

**Part I - Summary**

Reviewer #2: (No Response)

Reviewer #3: (No Response)

**Part II – Major Issues: Key Experiments Required for Acceptance**

Reviewer #2: (No Response)

Reviewer #3: (No Response)

**Part III – Minor Issues: Editorial and Data Presentation Modifications**

Reviewer #2: (No Response)

Reviewer #3: The revised manuscript satisfactorily deals with the reviewer concerns. It contains new data that increases the support for the conclusions of the paper, including basic data on the adherent lines and overexpression data of Bb10 and Bb60 in the BS-resistant line.

I have only a few minor points left that I recommend to consider:

- The data in Fig. S5 is important (without it the evidence for Bb10 and Bb60 in BSD transport is rather weak). Could it also be part of a main figure?

- + and - GlcN is mixed up in Fig S8

- one further explanation for what the authors call 'leakiness' of the glmS line could be that insertion of the glmS sequence in the 3'-region affected the stability or translation of the target mRNA independent of its interaction with GlcN.

- Please get the manuscript checked by a native speaker: Some examples:

line 174: Two proteins showed signals inside parasite... add 'the' before parasite

line 185: Remaining seven candidates did not show signals... add 'the' at start of sentence

line 363: inserted to the 3' end of the BbVEAP ORF the same as BbVEAP-myc-glmS line. 'the same way as'?

line 366: with GlcN treatment by Western blot analysis (S8A Fig). 'as determined by Western blot analysis'?

line 426: These observation... observation should be plural

line 429 the expression of other mtms were unchanged... 'was unchaged'

PLOS authors have the option to publish the peer review history of their article (what does this mean?). If published, this will include your full peer review and any attached files.

Reviewer #2: No

Reviewer #3: No
---

## [Editor Report · Decision Letter 2]

20 Aug 2020

Dear Dr. Asada,

We are pleased to inform you that your manuscript 'Novel Babesia bovis exported proteins that modify properties of infected red blood cells' has been provisionally accepted for publication in PLOS Pathogens.

Best regards,

David R. Allred, Ph.D.

Guest Editor

PLOS Pathogens

Kirk Deitsch

Section Editor

PLOS Pathogens

Kasturi Haldar

Editor-in-Chief

PLOS Pathogens

orcid.org/0000-0001-5065-158X

Michael Malim

Editor-in-Chief

PLOS Pathogens

orcid.org/0000-0002-7699-2064

This manuscript provides some very intriguing new characters in Babesia biology that are worthy of attention and followup.
---

## [Editor Report · Acceptance letter]

25 Sep 2020

Dear Dr. Asada,

We are delighted to inform you that your manuscript, "Novel Babesia bovis exported proteins that modify properties of infected red blood cells," has been formally accepted for publication in PLOS Pathogens.

Best regards,

Kasturi Haldar

Editor-in-Chief

PLOS Pathogens

orcid.org/0000-0001-5065-158X

Michael Malim

Editor-in-Chief

PLOS Pathogens

orcid.org/0000-0002-7699-2064